# Dissolving Activated Carbon Pellets for Ibuprofen Removal at Point-of-Entry

Michelle Finn *, Noelle Yackel, Gabrielle Giampietro and David Mazyck

Environmental Engineering Sciences, 314 AP Black Hall, University of Florida, Gainesville, FL 32611, USA; noelle.yackel@ufl.edu (N.Y.); giampietrog@ufl.edu (G.G.); dmazyck@ufl.edu (D.M.)
* Correspondence: mfinn2012@ufl.edu

**Abstract:** The increased usage of pharmaceuticals coupled with the desire for greywater reuse to reduce the freshwater demand for potable water requires a user-friendly engineered solution. Activated carbon is a proven technology that is typically used for organic pollutant removal at water treatment plants. Lignite, coconut, and a blend of activated carbon powders were used to develop rapid-dissolving pellets with an inorganic binder. Ibuprofen was the model compound chosen for pharmaceutical adsorption in deionized water and synthetic hydrolyzed and synthetic fresh urine at rapid contact times (0.5 to 30 min) and using various pellet dosages (0.5 to 10 g/L). A cost analysis was performed to determine the feasibility of the engineered solution. With an increase in contact time, the coconut pellets outperformed both the blend and lignite pellets in deionized water at a set pellet dosage. The lignite pellets were the most cost-effective with rapid adsorption in fresh urine and a capacity of 0.089 g ibuprofen/g pellet. Additional optimization parameters include pellet dissolvability, pellet dosage in relation to different pharmaceuticals, and the impact of activated carbon on the household sewage system, and each of these are necessary to determine application feasibility.

**Keywords:** activated carbon; ibuprofen; carbon pellet; water treatment

## 1. Introduction

Recent studies have shown that conventional wastewater treatment plants (WWTPs) perform poorly when removing pharmaceuticals from wastewater [1,2]. There is an increased demand worldwide for available drinking water. Several European countries have placed more stringent requirements on water reuse, focusing on pharmaceuticals and pesticides [1,2]. WWTPs are not designed to focus on pharmaceutical removal but on large organic molecules, such as natural organic matter; thus, a point-of-entry (POE) solution to prevent compounds from reaching WWTPs is attractive. A decrease of the effluent pharmaceutical concentration will reduce the potential to negatively impact the environment, drinking water supply, and human health.

There have been at least 700 compounds identified in European water that have the potential to cause adverse ecological and human health effects [3]. These compounds are introduced through anthropogenic activities that contribute to the contamination of the water supply with pharmaceuticals being included in the identification. More than 70% of ingested pharmaceuticals are excreted through human urine [4]. The pharmaceuticals found in waste streams include but are not limited to, non-steroidal and anti-inflammatories (e.g., ibuprofen, diclofenac, naproxen, and ketoprofen) which have been found in surface water at low concentrations (ng/L and μg/L) [5]. Adverse developmental patterns have been observed in aquatic species in the presence of low pharmaceutical concentrations of drugs such as ibuprofen, diclofenac, the antidepressant fluoxetine, and steroids such as estrogens and progestogens [6]. The negative effects observed include impacts on the reproductive system, changes in temperament, and organ damage. The toxicological impact on humans

after an extended period of low-concentration exposure is still unclear [7–10]. Some of these low-dosage impacts may include obesity, neurobehavioral disorders, infertility, and immune dysfunctions [10,11].

The application of POE and point-of-use (POU) treatment is especially prevalent in areas with limited potable water access. In northern New England, with more than 40% of the population relying on private wells as the primary drinking water source, fixed bed POE and POU treatment systems are implemented as an effective way to remove elevated levels of arsenic. Möller et al. [12] tested 275 POU and POE systems of hybrid hydrous iron oxide/polymer media for arsenic levels. They found that the initial breakthrough of the systems did not occur until after 1.5 years of operation. Such systems have also been successfully implemented in New Jersey because the local government took an aggressive approach to the state-wide arsenic maximum contaminant level (MCL) of 5 ppb (less than half of the federal MCL); this must be met by private customers and owners of small communities [13]. Legislation has been demonstrated to be an effective driving force for pollutant treatment.

Activated carbon is an existing air and water treatment technology implemented full-scale for pollutant removal. It has a high surface area (e.g., 500–1200 $m^2/g$) and can be obtained in the form of powder or granular or pelletized material [14–16]. Its application has been well-documented for organic and inorganic removal [17–19] and is the EPA's best available technology for the treatment of synthetic and volatile organic compounds.

A relatively novel approach to introducing AC as a POE application to a system is in the form of pellets or briquettes. Pelletized or briquette-activated carbon is produced by employing a binder with a powdered starting material. The organic binders (e.g., coal tar or sugar-containing waste) are typically employed with subsequent activation of the pellet [20]. The binder is expected to increase the strength of the pellet while also improving the porosity. A liquid binder may cause a decrease in the porosity initially by filling cracks and voids in the powdered carbon surface [21]; however, through subsequent activation, the pelletized material has been shown to increase in surface area [22]. The surface area and structural integrity are dictated by the binder-to-powdered material ratio. The authors have not found any publications detailing the dissolvability of pelletized AC material for the application of rapid aqueous-phase adsorption.

The research on activated carbon for ibuprofen adsorption focuses predominantly on surface functional groups and porosity modification. The basic surface area and accessible microporosity have been demonstrated to positively impact ibuprofen adsorption [23,24]. The carbon is primarily applied as a powder treatment option, but pelletized carbon has been researched for the aqueous phase. The pellets were extruded and designed to maintain their structure when in contact with an aqueous solution for the adsorption of organic compounds and have comparable adsorption to the PAC and the granular activated carbon materials used for comparison [25,26].

In this work, rapid dissolving activated carbon pellets were engineered for the adsorption of ibuprofen in the aqueous phase. The aim is to remove ibuprofen in both deionized water as a model system and in synthetic urine (fresh and hydrolyzed). The effects of the pellet dosage, carbon type, and contact time were studied.

## 2. Materials and Methods

### 2.1. Adsorbates

Ibuprofen sodium-salt (Sigma Aldrich, St. Louis, MO, USA) was used as the model pharmaceutical compound to quantify adsorption. Ibuprofen is a hydrophobic compound, requiring an alcohol to achieve complete dissolvability. Alcohol solutions will competitively adsorb to surface functional sites of activated carbon and are avoided for the current application [27,28]. Ibuprofen–sodium has a high solubility limit in water, which made it ideal for this research. A concentration of 1 g/L was chosen, unless otherwise noted, to model the concentration of approximately 25 ibuprofen pills (at 200 mg/pill) discarded into a standard 1.6 gallon (6.5 L/flush) toilet commonly used in the United States.

Solutions of synthetic fresh urine and synthetic hydrolyzed urine were created to replicate competitive adsorption within the toilet bowl effluent. The components of synthetic hydrolyzed urine and fresh urine were previously studied [29–31]. Table 1 contains the compositions of each solution.

**Table 1.** Components of synthetic hydrolyzed urine and fresh urine [29–31].

| Chemical Name | Supplier | Fresh Urine (mmol L$^{-1}$) | Fresh Urine (mg mL$^{-1}$) | Hydrolyzed Urine (mmol L$^{-1}$) | Hydrolyzed Urine (mg mL$^{-1}$) |
|---|---|---|---|---|---|
| Sodium Chloride | Fisher Scientific, Waltham, MA, USA | 44 | 2.57 | 60 | 3.51 |
| Sodium Sulfate | Fisher Scientific, Waltham, MA, USA | 15 | 2.13 | 15 | 2.13 |
| Potassium Chloride | Fisher Scientific, Waltham, MA, USA | 40 | 2.98 | 40 | 2.98 |
| Magnesium Chloride | Fisher Scientific, Waltham, MA, USA | 4 | 0.81 | - | - |
| Sodium Phosphate Dibasic Anhydrous | Acros Organics, Waltham, MA, USA | 20 | 0.62 | 5 | 0.16 |
| Calcium Chloride Dihydrate | Acros Organics, Waltham, MA, USA | 4 | 0.59 | - | - |
| Ibuprofen Sodium Salt | Sigma Aldrich, St. Louis, MO, USA | 2 | 0.50 | 2 | 0.50 |
| Sodium Citrate | Sigma Aldrich, St. Louis, MO, USA | 0.07 | 0.02 | 0.07 | 0.02 |
| Urea-N | Acros Organics, Waltham, MA, USA | 500 | 7.00 | - | - |
| Ammonium Hydroxide, ACS grade | JT Baker Avantor, Allentown, PA, USA | - | - | 233 | 15.70 mL |
| Ammonium Bicarbonate | Fisher Scientic, Waltham, MA, USA | - | - | 267 | 21.11 |

### 2.2. Activated Carbon Pelletization

Coconut and lignite powdered activated carbon (PAC) were used as adsorbents to create pelletized activated carbon. The PAC were sized to less than 45 μm and dried at 150 °C to remove moisture. The extrusion process utilized a Bonnet pelletizing extruder (Akron, OH, USA) with a 4 mm die plate attachment to create a 4 mm diameter and 4–6 mm length pellet. A mixture of water and calcium bentonite was combined with the activated carbon as a binder to create an extrudable material. Table 2 contains the wet pellet formulation.

**Table 2.** Wet 4 mm pellet formulation of P1-coco, P2-blend, and P3-lig.

| Pellet Sample ID | Pellet Formulation |
|---|---|
| P1-coco | 1.2 lb Coconut PAC and 0.3 lb inorganic binder |
| P2-blend | 0.6 lb Coconut PAC, 0.6 lb Lignite PAC, 0.3 lb inorganic binder |
| P3-lig | 1.2 lb Lignite PAC and 0.3 lb inorganic binder |

### 2.3. Sample Characterization

#### 2.3.1. BET Surface Area

Nitrogen adsorption/desorption in combination with the Quantachrome NOVA 2200e device (Boca Raton, FL, USA) was used to analyze the porosity characteristics of each pellet.

Prior to analysis, the samples were held at 110 °C and then put in a desiccator overnight. Ultra-high purity nitrogen gas (NexAir), kept at a constant temperature of −196 °C and submerged in a liquid nitrogen bath, was used as the adsorbate for the isotherm to analyze the activated carbons. The total pore volume was determined by plotting the volume of the adsorbed nitrogen gas versus the relative equilibrium pressure. In this procedure, it was assumed that all the pore spaces were filled with the adsorbate and acted under a limiting pressure of $P/P0 = 0.99$. The Brunauer–Emmett–Teller (BET) equation at a $P/P0 = 0.01–0.3$ was used to calculate the surface area of each sample and analyzed in duplicates. The slope and y-intercept provided the constant (C) of the isotherm. This 5-data-point BET calculation was validated with positive reoccurring C values.

### 2.3.2. Ash Content

The ash contents of the three pellets (P1-coco, P2-blend, and P3-lig) and PAC (lignite and coconut) were determined by the ASTM-D2866 [32] standard test method for the total ash content of activated carbon. The samples were sized accordingly before the test.

### 2.3.3. Pellet Dissolvability

The dissolvability test was conducted to determine the time required to dissolve each pellet sample. P1-coco, P2-blend, and P3-lig were sieved from 2.8 to 3.35 mm and weighed to approximately one gram. The dry and wet masses of the 2.0 mm by 2.0 mm mesh basket were taken. The pellet samples were then placed into the basket and submerged in deionized water at a marked depth in a 1.0 L jar tester rotating at 60 RPM. The mass of the basket and pellets was recorded at 30-s increments until no pellets remained in the basket. The test was performed in triplicate.

### 2.3.4. Sample Density

The density of the activated carbon pellets and powders was determined by ASTM D2854 and D8176, respectively [33,34]. All the samples were dried at 150 °C and cooled in a desiccator to reduce the moisture adsorbed. The pellet density was determined by sieving the pellets (2.80–3.35 mm) (Gilson, Lewis Center, OH, USA) and placing them into the vibratory feeder (Syntron, Saltillo, MS, USA) at a 4.1 power pulse to slowly fill the graduated cylinder. The resulting volume and weight were recorded to determine the pellet density.

The mechanically tapped density of powdered activated carbon (PAC) was measured using an Autotap (Quantachrome, Boca Raton, FL, USA). A 100 mL graduated cylinder was sealed and placed on the device for 30 min. The resulting volume and weight were recorded to determine the powder density.

### *2.4. Analytical Method and Calibration*

The concentrations of ibuprofen–sodium were analyzed on an ultraviolet-visible (UV) spectrophotometer (Hach DR 6000) using a 1 cm quartz cuvette. The following methods were adapted from previous literature, which tested biochar for ibuprofen adsorption using UV spectroscopy [30]. The initial wavelength scans were obtained from 200–800 nm to observe the peak absorbance of ibuprofen. This was used to calculate the unknown concentration during the batch adsorption experiments. A calibration curve was developed for ibuprofen–sodium in deionized water with increasing concentrations from 0.0, 50.0, 250.0, 500.0, 750.0, and 1000.0 mg/L at 224 nm ($y = 1.3484x + 0.0019$). The coefficient of determination ($R^2$) value was 0.9990 (Figure 1).

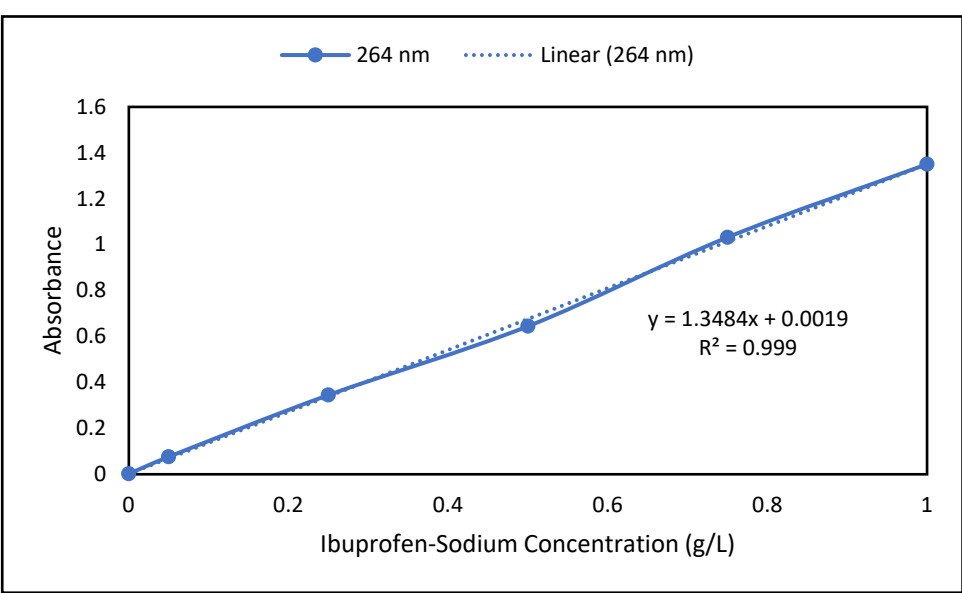

**Figure 1.** Ibuprofen–sodium calibration from 0 to 1.0 g/L in deionized water using a 1 cm quartz cuvette at 264 nm wavelength.

Synthetic fresh and hydrolyzed urine at 500.0 mg/L ibuprofen–sodium concentration wavelength scans were compared with 500.0 mg/L ibuprofen–sodium in deionized water to determine whether a wavelength shift or peak increase occurred from the additional compounds in the solution. A total of 500.0 mg/L was chosen, as this is the stock concentration used to determine the removal of pharmaceuticals in synthetic urine in this study. Figure 2 demonstrates the wavelength scans of the solutions with no shift in the peak wavelength or changes in absorbance at 264 nm.

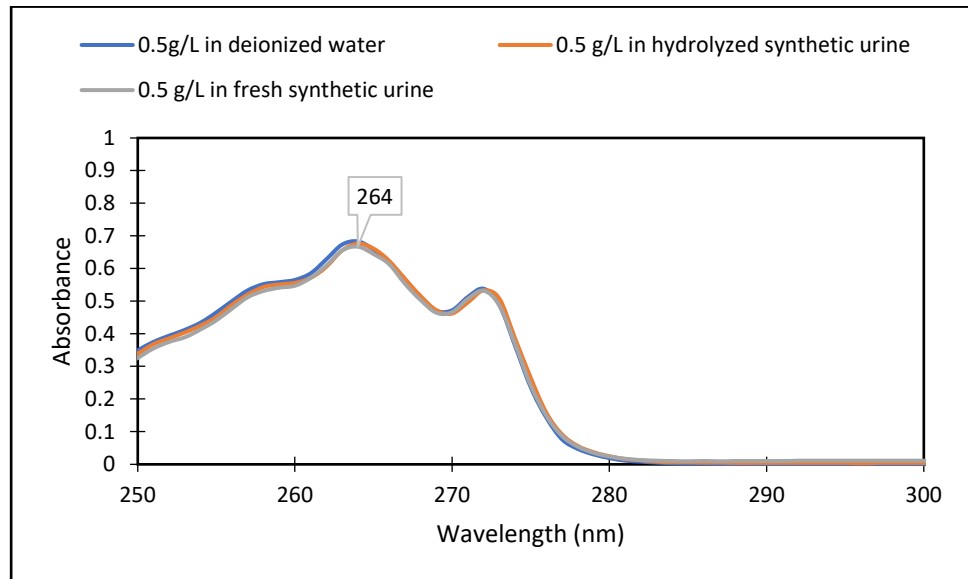

**Figure 2.** Ibuprofen–sodium wavelength spectrum at 0.5 g/L in deionized water, hydrolyzed synthetic urine, and fresh synthetic urine water using a 1 cm quartz cuvette to demonstrate no peak shift at 264 nm.

### 2.5. Batch Adsorption

Batch adsorption experiments were conducted with the lab-made pellets P1-coco, P2-blend, and P3-lig in various solutions to evaluate the adsorption of ibuprofen sodium salt (Figure 3). Deionized water, fresh urine, and hydrolyzed urine were used to simulate

potential ibuprofen removal within a toilet bowl. The pellet masses varied from 1.0–5.0 g/L in DI water, synthetic fresh urine, and synthetic hydrolyzed urine at contact times of 30 to 120 s. Short contact times were selected to mimic the contact time in the toilet bowl prior to solution introduction to the main household sewer system. Additional contact times (1 to 30 min) and pellet dosages were tested to determine the minimum requirement to achieve at least a 50% reduction in the ibuprofen concentration.

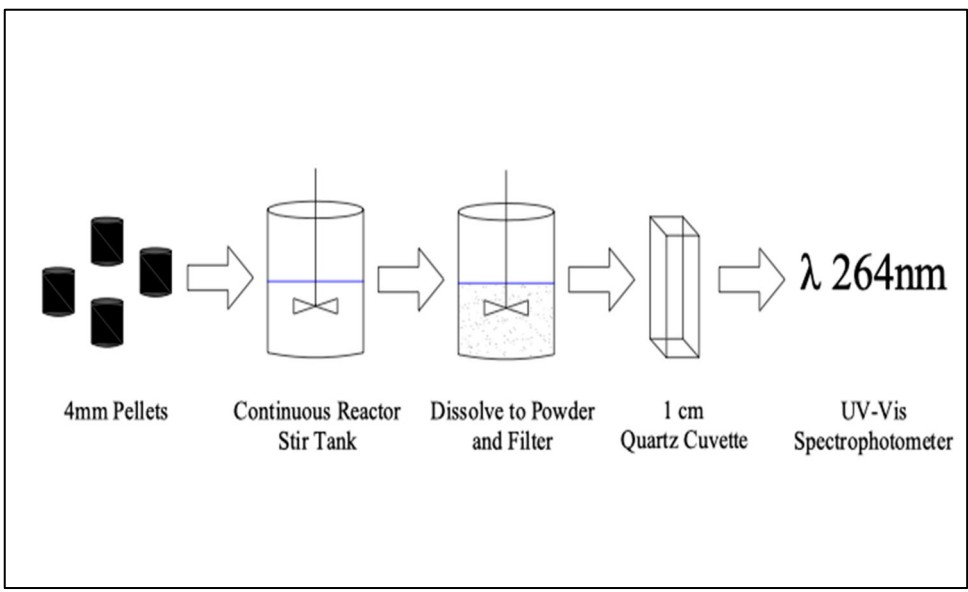

**Figure 3.** Batch adsorption schematic of activated carbon pellets for ibuprofen analysis. Black cubes demonstrate the pelletized material. The blue line represents the water line where the pellets are dissolved into a powder.

The pellets were sieved from 2.8 to 3.35 mm and stirred in a solution at 60 RPM. Aliquots were filtered using a syringe and a 0.45 μm nitrocellulose filter. The final ibuprofen concentration of each sample was analyzed using a UV spectrophotometer at 264 nm.

## 3. Results and Discussion

Lignite- and coconut-derived powdered activated carbon (PAC) were utilized as the base material for the pelletized technology. The PAC were analyzed for their physical and chemical properties (Table 3). The lignite PAC contains a high initial ash content and has a basic surface chemistry (pH 10.5). This is expected of lower-ranked coal [35]. The coconut PAC is microporous with a higher BET surface area of 1088 $m^2$/g compared to the lignite PAC of 509 $m^2$/g. Both commercially available materials are thermally activated at the commercial level to increase the surface area; it is expected that basic oxygen surface functional groups primarily populate the carbon surface.

**Table 3.** Physical and chemical properties of the PAC samples.

| Powdered Sample ID | Tapped Density (g/mL) | Ash Content (%) | Contact pH | BET ($m^2$/g) | Average Pore Size (Å) | Cumulative Pore Volume (cc/g) | Meso-Pore Volume (cc/g) | Micro-Pore Volume (cc/g) |
|---|---|---|---|---|---|---|---|---|
| Lignite PAC | 0.51 | 28.5 | 10.5 | 509 | 37.1 | 0.472 | 0.248 | 0.152 |
| Coconut PAC | 0.93 | 1.8 | 9.7 | 1088 | 17.8 | 0.483 | 0.036 | 0.388 |

The PACs were further engineered in a pelletized form by creating a dry material blend (without water) of 80% coconut/20% inorganic binder (P1-coco), 40% coconut/40% lignite/20% inorganic binder (P2-blend), and 80% lignite/20% inorganic binder (P3-lig). The formula makeups utilized are based on preliminary data for optimal adsorption, pellet

dissolvability, and ease of production. The notable impacts of pelletization are an increase in the ash content and a decrease in contact pH, both deriving from the inorganic binder (Table 4). Pelletizing the powder decreased the total surface area by 14% and 10% for P1-coco and P3-lig, respectively. The pellet porosity is not further discussed in this article, as the pellets will reduce in particle size back to powders (<45 μm) in a solution.

**Table 4.** Physical and chemical properties of the engineered pellets.

| Pellet Sample ID | Apparent Density (g/mL) | Ash Content (%) | Contact pH | BET (m²/g) | Average Pore Size (Å) | Cumulative Pore Volume (cc/g) | Meso-Pore Volume (cc/g) | Micro-Pore Volume (cc/g) |
|---|---|---|---|---|---|---|---|---|
| P1-coco | 0.45 | 19.4 | 8.6 | 933 | 19.1 | 0.445 | 0.060 | 0.332 |
| P2-blend | 0.50 | 39.8 | 7.4 | 606 | 25.7 | 0.390 | 0.131 | 0.204 |
| P3-lig | 0.48 | 42.4 | 8.9 | 456 | 35.9 | 0.409 | 0.205 | 0.136 |

*3.1. Pellet Dissolvability-Basket Test*

The pellet samples were submerged in deionized water in a 1 mm metal mesh and 27.5 cm length basket (O.D. 3.5 cm) to determine how the sample would disintegrate in turbulent water. A basket was used to contain the pellets and reduce sample disintegration caused by pellet collisions. The sample formulations were primarily optimized for the highest mass of activated carbon to binder ratio and not for dissolvability. The inorganic binder, which swells upon contact with water, allows for rapid disintegration.

The P1-coco lost its pellet structure and dissolved to resemble a powder within 150 s. The P2-blend and P3-lig dissolved in less than 30 s. The impact of particle size was previously demonstrated by Finn et al. [16] to show that overcoming intraparticle diffusion is necessary for the adsorption of ibuprofen. Therefore, rapid adsorption is expected to be the greatest in the P2-blend and P3-lig due to the faster reduction of the particle size. The reasoning behind P1-coco requiring fivefold more time than the P2-blend and P3-lig was not explored further but may involve the hydrophilic nature of the lignite carbon [36]. Hydrophilicity occurs from oxygen functional groups, such as aldehydes, alcohols, carbonyls, carboxyls, etc., and can contribute to rapid pellet dissolution [37].

*3.2. Batch Adsorption-Deionized Water and Synthetic Urine Solutions*

Batch adsorption at various rapid contact times was performed to investigate the kinetics first in deionized water d, then in synthetic urine solutions. Ibuprofen is water-insoluble due to the presence of non-polar alkyl groups and benzene rings; therefore, ibuprofen–sodium was utilized to ensure complete solubility without potential competitive adsorption from a solvent [28]. It will exist as a weak acid in a solution with a pKa of 4.4 [38].

In a standard toilet system built after 1992 containing ~6.06 L (1.6 gallons), 30 pills (200 mg ibuprofen) would need to be added to equate to roughly 1 g/L concentration [39]. The work focuses on adsorption using pure ibuprofen salt rather than ibuprofen compressed in a pill; competitive adsorption is expected to occur between ibuprofen and the inactive ingredients (e.g., binder and coloring dye) in a pill to the carbon surface. This will require further experimentation to understand the adsorption parameters. Competitive adsorption may be difficult to predict, as conventional drug delivery can involve tablets, gels, and capsules that require different binders based on the desired solubility, i.e., fat-soluble or water-soluble. Therefore, a model compound is helpful in comparing various removal technologies.

For all the analyses, the measured initial concentration of the solution was 0.96 g/L. The experiments were carried out without a pH adjustment. Figure 4 demonstrates the ibuprofen percent removal at 15 min of contact time using various pellet dosages. As the pellet dosage increased, the removal increased. P1-coco maintained a higher percent removal than P3-lig at all pellet dosages. At 10.0 g/L, the P1-coco and P2-blend had comparable percent removals at 87% and 88% removal, respectively.

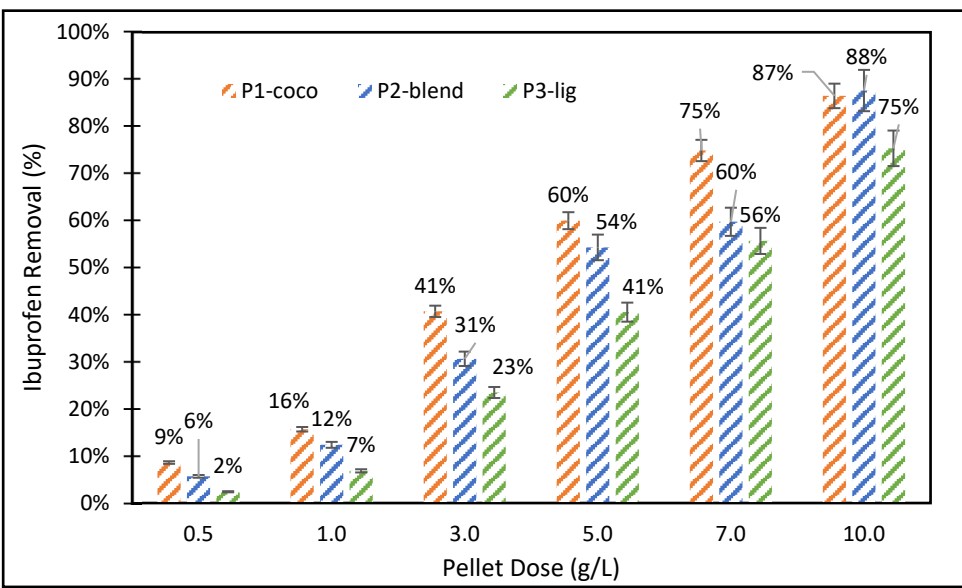

**Figure 4.** Ibuprofen percent removal at 15 min contact for pellet samples at various dosages.

The removal of ibuprofen at POE in wastewater occurs under a short contact time prior to the introduction of additional competitive species, primarily natural organic matter. Therefore, the kinetic adsorption was investigated between the range of 1 to 30 min, and the results are presented in Figure 5. At one minute, the P2-blend and P3-lig adsorbed 34% and 33% ibuprofen, respectively. The P1-coco had minimal removal of 22% at 1 min, but this significantly increased with contact time. At five minutes, the P1-coco had the highest removal at 53%, followed by the P2-blend and P3-lig at 43% and 37%, respectively. For all the samples, removal achieved equilibrium at 30 min of 61%, 51%, and 40% for P1-coco, P2-blend, and P3-lig, respectively.

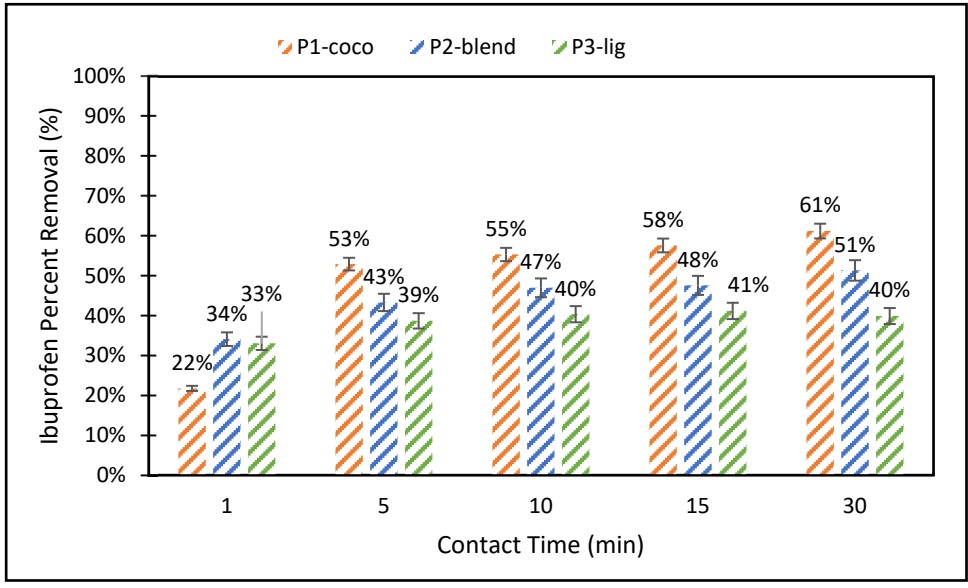

**Figure 5.** Effect of contact time on ibuprofen adsorption for various pellet samples at 25 °C (Ci = 1.0 g/L ibuprofen, pellet dosage = 5.0 g/L).

The analysis of ibuprofen in fresh and aged (hydrolyzed) urine was completed to determine removal at the point-of-source (fresh) or aged urine (hydrolyzed). The treatment of aged urine is assumed to be the ubiquitous solution occurring for wastewater. The composition of urea will hydrolyze to ammonia and bicarbonate with an increase in pH;

fresh urine will maintain a pH of 6, while the hydrolyzed urine will have a pH of 9 [31]. Additionally, the hydrolyzed urine is expected to have a higher ionic strength, which will impact the adsorption [40,41].

The pelletized samples were analyzed at rapid contact times of 30, 60, 90, and 120 s at an initial concentration of 0.5 g/L ibuprofen and at various pellet doses with minimal competition (deionized) and fresh/hydrolyzed urine. Figure 6 presents the ibuprofen percent removal of P1-coco as the adsorbent. Competitive adsorption is most apparent in the hydrolyzed urine solution with less than 15% removal at 120 s for 5 g/L of pellets.

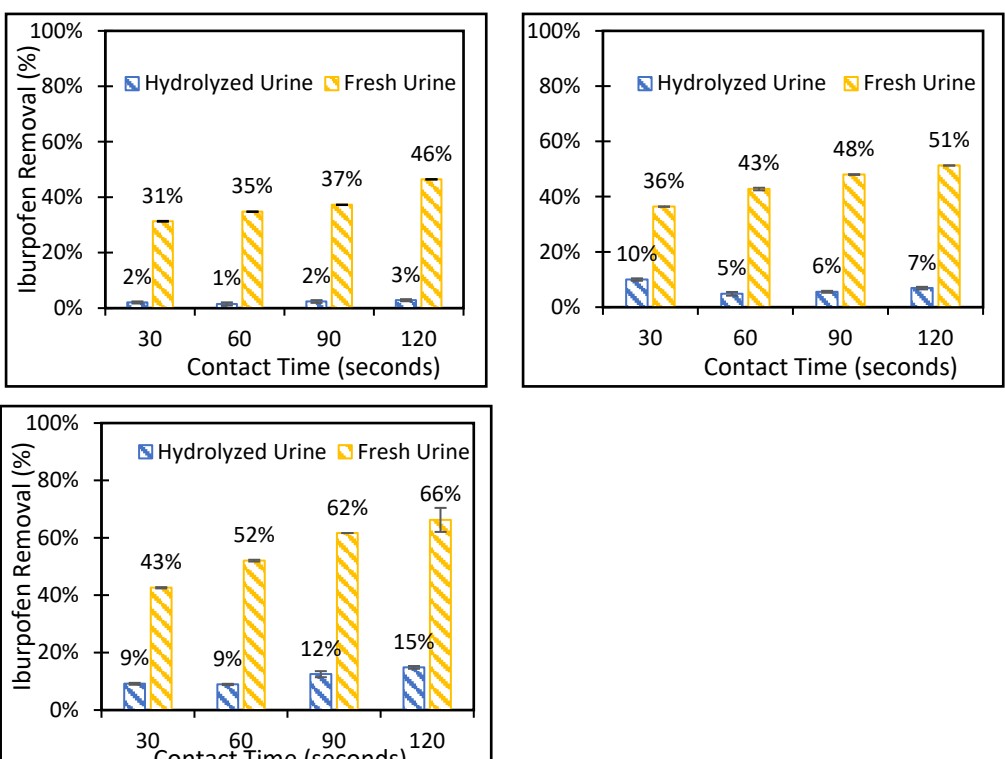

**Figure 6.** Ibuprofen percent removal using sample P1-coco in hydrolyzed and fresh urine at short contact times of 30–120 s at various dosages of 1 g/L (**top left**), 3 g/L (**top right**), and 5 g/L (**bottom left**).

The coconut AC demonstrated a decrease in the surface area and microporous volume as pellets, as can be seen in Table 3; there is an increase in the mesoporous volume, allowing for a decrease in pore blockage that typically occurs in very microporous carbons [42]. However, the pellet porosity will reduce to the original powder porosity as it dissolves in water. The pellet was reported to require at most 150 s to achieve complete dissolution in deionized water without factoring in pellet collision. However, pellet collision is expected to increase the possibility of disintegration.

The competitive adsorption of the blended pellet (P2-blend) demonstrates an increase in the percent removal as the pellet dosage and contact time increase, as seen in Figure 7. The pellet disintegrated in less than 30 s, similar to the P3-lig, allowing for a reduction in the particle size distribution and thus improving the adsorption kinetics. Ibuprofen adsorption is most prominent in the fresh urine, compared to the hydrolyzed urine. This is a good indication that the pellets will be successful for POE removal with a short contact time. The increase in the removal of ibuprofen from fresh urine was from 37% to 40% and 46% to 51% for 1 g/L and 3 g/L pellet dosage, respectively. There was a larger increase in the removal of ibuprofen from fresh urine using 5 g/L pellet dosage, from 54% to 68%.

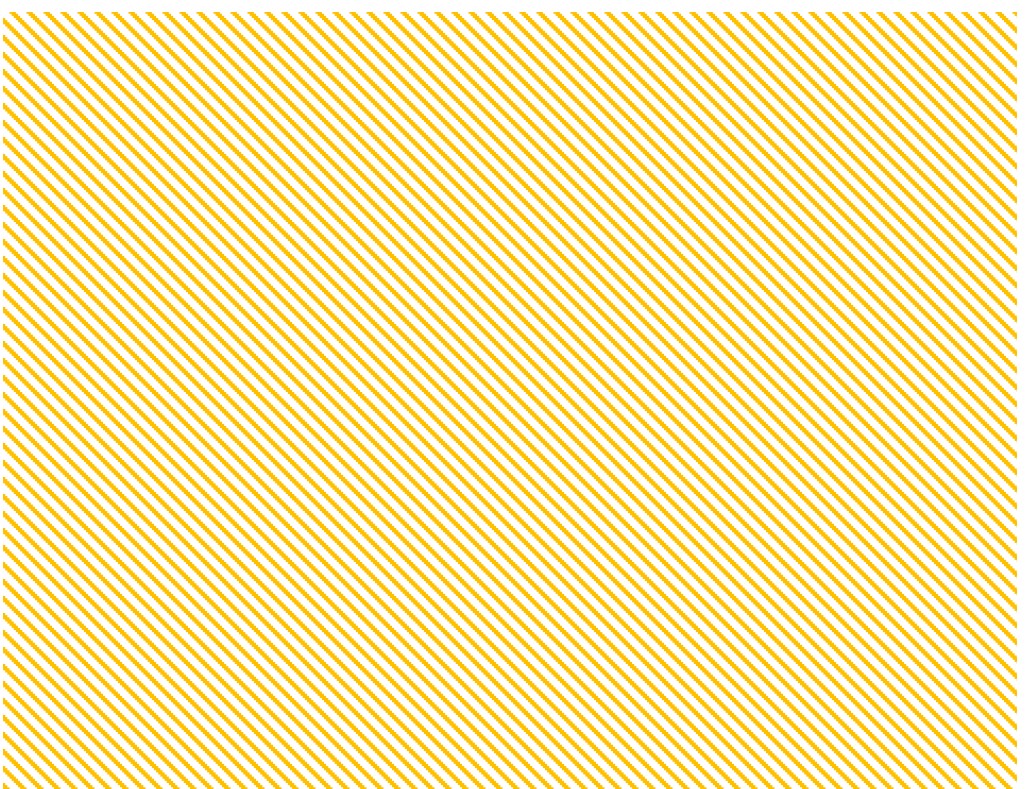

**Figure 7.** Ibuprofen percent removal using sample P2-blend in hydrolyzed and fresh urine at short contact times of 30–120 s at various dosages of 1 g/L (**top left**), 3 g/L (**top right**), and 5 g/L (**bottom left**).

The P2-blend is comprised of 50% microporous carbon and 50% mesoporous carbon. It shows comparable ibuprofen removal to P1-coco in fresh urine but greater removal in hydrolyzed urine. The blend of activated carbons offers both a microporous and mesoporous carbon available for adsorption. Blended activated carbon material has been previously studied for pesticide removal, the oil and fat refining process, and dye adsorption to obtain the desired porosity and adsorption capacity while reducing costs (promoting higher overall carbon yield and reducing the burn-off percentage) [43,44]. Therefore, there is a cost-benefit to engineering a blended sample to optimize performance while reducing production costs. However, there may be difficulty in understanding the mechanisms, as two different materials are being applied simultaneously.

P3-lig demonstrates the highest ibuprofen removal in hydrolyzed urine at 11%, 24%, and 35% at 30 s of contact time for a dosage of 1 g/L, 3 g/L, and 5 g/L, respectively (Figure 8). The percent removal does not significantly increase with an increase of time. The percent increase is +3%, +8%, and +4% for the various pellet dosages from 30 to 120 s in fresh urine. The most significant change in removal was found using 5 g/L P3-lig in hydrolyzed urine with 35% removal at 30 s and 58% removal at 120 s.

The lignite PAC surface area is less than half of the coconut PAC surface area, while demonstrating higher initial ibuprofen removal at 30 s. It is also worth noting that the P3-lig pellet completely dissolved to a powder at 30 s. P3-lig has a greater mesopore volume with less micropore volume in comparison to the P1-coco and P2-blend, which is expected to aid in the competitive adsorption of ibuprofen. As a powder, lignite PAC has a similar cumulative pore volume to coconut PAC of 0.472 cc/g (with coconut being 0.482 cc/g). The difference remains in the mesopore vs. micropore volume. Lignite PAC has a higher mesopore volume of 0.248 cc/g compared to 0.036 cc/g for coconut PAC (Table 3).

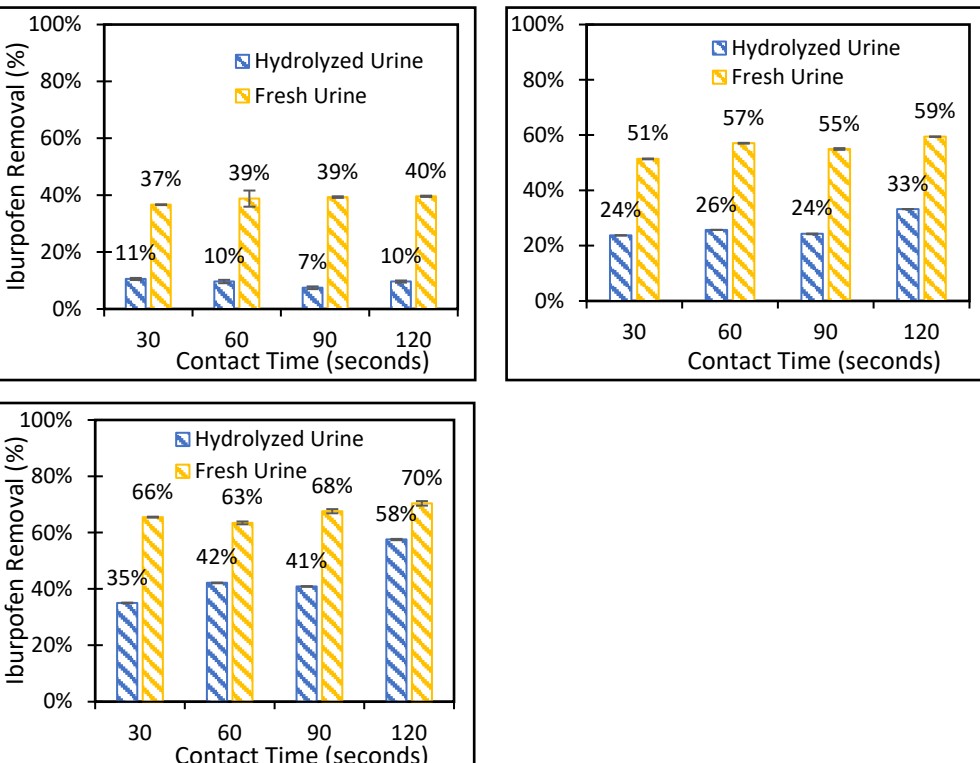

**Figure 8.** Ibuprofen percent removal using sample P3-lig in hydrolyzed and fresh urine at short contact times of 30–120 s at various dosages of 1 g/L (**top left**), 3 g/L (**top right**), and 5 g/L (**bottom left**).

### 3.3. Cost-Benefit Analysis

A cost-benefit analysis of the engineered technology is important to determine the application feasibility. For POE removal, it can be assumed that equilibrium is not achieved prior to the carbon–ibuprofen–urine mixture entering the household drainage system. Therefore, kinetics should be compared. Table 5 contains the adsorption performance in fresh urine and deionized water at 60 s contact time and a pellet dosage of 5 g/L. The low-cost binder, similar to silica–alumina, carboxymethyl cellulose, starch, or dextrin material, is estimated to be USD 0.10/lb. The activated carbon raw material prices of USD 0.90 for coconut and USD 0.65 for lignite is an average estimated price and does not factor in transportation, tariffs, and market fluctuation [45–47].

**Table 5.** Cost-benefit analysis of P1-coco, P2-blend, and P3-lig comparing cost to quantified ibuprofen capacity at 60 s.

| Pellet ID | Raw Material Cost ($/lb) | Total Pellet Cost ($/lb) | $Q_{1\ min-fresh\ urine}$ (g Ibuprofen/g Pellet) | $Q_{1\ min-deionized\ water}$ (g Ibuprofen/g Pellet) |
|---|---|---|---|---|
| P1-coco | 0.90 | 0.74 | 0.070 | 0.042 |
| P2-blend | 0.78 | 0.64 | 0.088 | 0.066 |
| P3-lig | 0.65 | 0.54 | 0.089 | 0.065 |

Based on the total estimated pellet cost and capacity, the preferred pellet is P3-lig for fresh urine and deionized water. The ibuprofen capacity at 60 s contact time in fresh urine for the pellet samples is 0.070, 0.088, and 0.089 g/g pellet for the P1-coco, P2-blend, and P3-lig, respectively (Table 5).

With the depletion of coal reserves [48], focusing long-term efforts on optimizing a coconut or wood raw material may be more beneficial. The primary advantage of using

coconut over coal is the decreased emission of greenhouse and acid gas [49]. Therefore, sustainable alternatives, such as agriculture byproducts, should be further pursued.

## 4. Conclusions

An engineered solution 4 mm × 4 mm dissolving pellets using coconut, lignite, and a blend was analyzed for rapid ibuprofen removal as a point-of-entry technology. The pellets were extruded with an inorganic binder, which swells upon contact with water, allowing the structure to dissolve back into a powder. The pellet formulation of 80% carbon/20% binder was used to allow for maximum carbon content and minimal binder, while maintaining the pellet structure. The samples were tested in deionized water and fresh and aged (hydrolyzed) synthetic urine at rapid contact times.

It was demonstrated that the P3-lig was the most cost-effective with rapid adsorption of 0.089 g ibuprofen/g pellet in fresh urine. With an increase in the contact time, the P1-coco outperformed both the blend and lignite pellet in deionized water; however, a comparison of rapid adsorption (less than 1 min) is more realistic.

With water reuse becoming a more desirable option, as it reduces the freshwater demand, point-of-source wastewater treatment options need to be investigated. Additional optimization of the research for more rapid pellet dissolvability, quantifying pellet dosage for different pharmaceuticals, and life cycle assessment of the pellets on household sewage systems are necessary to determine the application feasibility.

**Author Contributions:** Conceptualization, M.F. and D.M.; methodology, M.F. and D.M.; validation, M.F. and N.Y.; formal analysis, M.F. and N.Y.; data curation, M.F. and N.Y.; writing—original draft preparation, M.F., N.Y. and G.G.; writing—review and editing, M.F., N.Y., G.G. and D.M.; visualization, M.F. and G.G.; supervision, D.M. All authors have read and agreed to the published version of the manuscript.

**Funding:** This research received no external funding.

**Institutional Review Board Statement:** Not applicable to study.

**Informed Consent Statement:** Not applicable to study.

**Data Availability Statement:** Research data is available upon email request to the corresponding author.

**Acknowledgments:** We acknowledge the support of Dan Ominski for acquiring the materials needed for experimentation and Leverto Jean Charles for aiding in the experimental analysis. We would also like to thank Carbonxt for providing the necessary equipment and space.

**Conflicts of Interest:** The authors declare no conflict of interest.

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
