# Peer review of "Dissolving Activated Carbon Pellets for Ibuprofen Removal at Point-of-Entry"

_processes, doi:10.3390/pr11051470_

Round 1

Reviewer 1 Report

This manuscript is interesting as its object is the removal of a compound such as ibuprofen, however the analytical method of measure the concentration of this pharmaucetical is not appropriate. To measure the concentration of ibuprofen is necessary HPLC analysis

Author Response

Reviewer 1, thank you for reviewing and providing comments. Please see the attached file where we address these comments.

Reviewer 2 Report

The authors submitted “Dissolving Activated Carbon Pellets for Ibuprofen Removal at Point-of-Entry.” to publish in “Processes”. The novelty and the quality of this work are good to publish in this journal. Here are my comments:

  1. The authors should peovide the TOC of this work.
  2. The authors should provide the schematic cartoon for the manuscript idea as a Figure 1.
  3. The authors did not see any BET data in this work.
  4. What about the morphology of this work?

Author Response

Reviewer 2, thank you for your comments. The manuscript has undergone extensive editing to address the comments made. Please see the attachment.

Reviewer 3 Report

The use of activated carbon pellets to remove ibuprofen was demonstrated in this research. Adsorption onto pellets is not often described in the literature, therefore this study is unique in that regard. The reviewer has some comments for the authors.

1. The use of UV-vis for ibuprofen analysis concerns me.  Human urine is a complicated matrix that could contain species that absorb UV at 264 nm.  What method did the authors generate calibration curves?  Was standard addition considered? LC-UV is recommended for complex samples like this. 

2. How many times these experiments were repeated? I do not see any error bars for these. I suggest the authors to include the error bars on the figures

3. It is obvious that as the dosage of adsorbent is increased, the removal percentage increases, but the capacity decreases, which must be addressed.

4. I found numerous typos and grammatical issues throughout the article, which must be corrected before it is considered for publishing.

Author Response

Reviewer 3, thank you for your comments. The manuscript has undergone extensive editing to address your comments. Please see the attachment.

Round 2

Reviewer 1 Report

The appropriate corrections have been made and the paper can be published

Reviewer 2 Report

This manuscript could be published in this journal.